

# Research and application of small-diameter hydraulic fracturing in-situ stress measurement system

Yimin Liu[1,2], Mian Zhang[2], Yixuan Li[3], Shuai Lu[2], Huan Chen[4,5*]

[1]Chengdu Technological University, Chengdu, 611730, China
[2]Tianjin Key Laboratory for Advanced Mechatronic System Design and Intelligent Control, School of Mechanical Engineering, Tianjin, 300384, China
[3]Chengdu Digital City Operation Management CO., LTD, Chengdu, 610041, China
[4]Institute of Exploration Technology, CGS, Chengdu, 611734, China
[5]Center for Geological Hazard Risk Prevention and Control Engineering Technology Innovation, Ministry of Natural Resources, Chengdu, 611734, China

*Correspondence to*: Huan Chen (77302709@qq.com)

**Abstract.** Observation and estimation of the stress state in the deep crust is a crucial challenge in in-situ stress measurement work. Hydraulic fracturing method is an important borehole-based technique for absolute in-situ stress measurement. The small-diameter hydraulic fracturing in-situ stress measurement system described in this article consists mainly of underground measurement components (serial small-diameter packers and dual-circuit connecting installation rods) and surface control components (hydraulic fluid control system, data acquisition system, and high-pressure oil pump with controllable flow). It enables series measurement of small-sized boreholes for in-situ stress, provides a maximum measurement range of 30~45 MPa. The subsequent calculation of in-situ stress data adopts the uniform design method to discuss the influence of various external factors on rock fracturing values. The small-diameter hydraulic fracturing in-situ stress measurement system has the advantages of simple and lightweight structure, short testing time, high success rate, and low requirements for rock integrity and pressurization equipment. It has formed a series of small-diameter in-situ stress measurement equipment, which has been innovatively promoted to the field of underground tunnel safety assessment in coal mines and metal mining areas. It has important practical value and economic significance in accurately determining the in-situ stress state of deep development areas.

.

## 1 Introduction

The stress that is undisturbed within the rock mass and exists in situ is referred to as in-situ stress. In-situ stress originates from multiple sources and is influenced by various factors, thus leading to a complex and variable distribution of stress within the crustal rock mass (Amadei and Stephansson, 1997). With the increasing demand for energy and mineral resources and the intensification of mining activities, both domestic and international mineral resources have entered a state of deep exploitation. However, the "three-high" problems (high ground stress, high ground temperature, and high water pressure)



encountered in deep mining have become the focal and challenging issues in rock mechanics research for deep mining (Xie, 2019).Accurately determining the in-situ stress state in the deep development spatial zone is one necessary approach to addressing the aforementioned challenges, which requires research on methods and techniques for in-situ stress test.

The hydraulic fracturing method is an important drilling-based technique for measuring absolute in-situ stress (Clark, 1949). In 1957, Hubbert and Rubey proposed that the fractures generated by hydraulic fracturing in boreholes are closely related to the existing stress state within the rock mass  (Hubbert and Rubey, 1959). Haimson and Fairhurst conducted theoretical and experimental analyses of the hydraulic fracturing in-situ stress testing technique in 1967, laying the theoretical foundation for the classical hydraulic fracturing testing technique (Haimson, 1968; Haimson and Fairhurst, 1967). In 1972, Von

Schonfeldt and Fairhurst carried out the first engineering practice of hydraulic fracturing in a borehole within an underground granite mass in Minnesota, USA. Since then, the hydraulic fracturing method for in-situ stress measurement has been widely applied in industries such as coal mine surrounding rock, hydropower stations, bridge tunnels, and cavern blasting. Moreover, the International Society for Rock Mechanics (ISRM) has recommended the hydraulic fracturing in-situ stress measurement method as a primary stress testing or estimation method for over 20 years.

After decades of development, domestic and international hydraulic fracturing in-situ stress measurement devices can mainly be classified into five categories (Wang, 2014), with their measurement methods and technical parameters shown in Table 1.

**Table1.** List of Hydraulic Fracturing In-situ Stress Measurement Devices

| No. | Measurement method | Research and development/manufacturer | Applicable borehole diameter (mm) |
|---|---|---|---|
| 1 | Cable-type | Professor Rummel from Germany | 70~110 |
| 2 | Light type | Australia CSIRO& Japan OYO | 45, 76 |
| 3 | Combined type | Professor Cornet and Thiercelin from France | 70~90 |
| 4 | BABHY type | Professor Ito from Japan | 48 |
| 5 | Drill rod type | Institute of Crustal Dynamics (CEA), Institute of Geomechanics (CGS), and Yangtze River Academy of Sciences, et.al | 50~90 |

Through literature research and market surveys, it is currently found that there is no hydraulic fracturing in-situ stress

measurement system with a diameter smaller than 45mm. However, the development of in-situ stress measurement equipment should aim for lightweight, simplicity, and flexibility. The series of small-diameter hydraulic fracturing in-situ stress measurement systems developed in this study have advantages such as lightweight equipment, simple structure, and low requirements for pressurization equipment. They can be pressurized using a manually operated system with high output pressure and low flow rate or a two-phase electric high-pressure oil pump, thereby reducing the overall weight and

manufacturing costs of the equipment. Additionally, the small-diameter measurement equipment can provide more abundant



data within the same borehole, greatly reducing the observation cost. Furthermore, it has lower requirements for intact rock mass, and by reducing the scale of rock integrity requirements, the measurement data becomes more accurate and reliable. Moreover, it enables the measurement of absolute stress in larger rock blocks within the fracture zone.

## 2 Testing theory of the in-situ stress

In hydraulic fracturing in-situ stress measurement, the drillholes are generally vertical and are primarily influenced by the maximum horizontal principal stress and the minimum horizontal principal stress. Therefore, the fracturing cracks are perpendicular to the plane of the minimum horizontal principal stress, resulting in vertical cracks (Zhang, 2018). There are two main classical mechanical models for fracture propagation in hydraulic fracturing: Khristinaovic-Geertsma-de Klerk (KDG) model (Geertsma and Klerk, 1969) and the Perkins-Kern-Nordgren (PKN) model (Perkins and Kern ,1961; Nordgren,

1972). The small-diameter measurement system described in this study adopts the PKN model to calculate the rock fracture value and combines it with the tensile strength of the core to infer the magnitude of the in-situ stress.

### 2.1 PKN mechanical model

The PKN model is based on the following assumptions: the rock is an elastic and brittle material, the height of the fracturing crack is constant, the cross-section of the crack is elliptical with the maximum crack width in the middle of the crack, and the

fracture toughness has no effect on the geometric deformation of the fracture (Wang et al., 2017). The fracture model of PKN is shown in Figure 1.

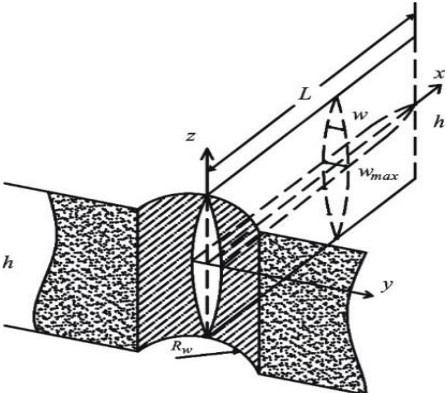

**Figure 1: PKN fracture model.**

Under the assumption of neglecting the compressibility of the fracturing fluid, Nordgren (1972) derived the fluid continuity

equation for the fracturing fluid in the crack as shown in Eq.(1).

$$\frac{\partial q}{\partial x} + q_t + \frac{\partial q}{\partial t} = 0 \tag{1}$$


In Equation 1, q(x , t) represents the volumetric flow rate of fluid through the cross-section of the crack; $q_t(x, t)$ represents the volumetric flow rate of fluid lost per unit length of the crack; A(x , t) represents the cross-sectional area of the crack. The calculation formulas for the crack length L, local fracture width w, and pore pressure $p_w$ in the absence of fluid loss are as

follows (Nordgren, 1972).

$$L = 0.68 \left[ \frac{GQ^3}{(1-\upsilon)\mu h^4} \right]^{\frac{1}{5}} t^{\frac{4}{5}} \tag{2-5}$$

$$w = 2.5 \left[ \frac{(1-\upsilon)\mu Q^2}{Gh} \right]^{\frac{1}{5}} t^{\frac{1}{5}} \tag{2-6}$$

$$p_w = 2.5 \left[ \frac{G^4 \mu Q^2}{(1-\upsilon)^4 h^6} \right]^{\frac{1}{5}} t^{\frac{1}{5}} \tag{2-7}$$

In Eq.(2), (3), and (4), G represents the shear modulus of the rock, ν represents the Poisson's ratio of the rock, h represents

the length of the crack, Q represents the fluid injection rate, and μ represents the viscosity of the fracturing fluid.

### 2.2 Principle of hydraulic fracturing method

The basic principle of in-situ stress measurement based on hydraulic fracturing involves placing drill rods and packers into a borehole using a drilling rig to measure their positions. Fluid is injected into the packers through a loading control system, isolating a test section within the borehole, and the fluid is further injected into the test section until fracturing occurs.

As shown in Figure 2, the first highest pressure value is recorded as the fracturing pressure $P_b$. Then the pressure drops rapidly to a state of fluid seepage into the fracture and remains constant. At this point, the pump is turned off to stop loading, and the pressure in the fracturing section decreases rapidly, causing the fracture to close quickly. When the fracture is in the near-closed state, the rate of pressure decrease slows down, and the pressure value at this time is recorded as the instantaneous closure pressure $P_s$. After releasing the pressure, reloading causes the fracture to reopen, and the pressure value

at this time is recorded as the reopening pressure $P_r$ (Wang et al., 2017).

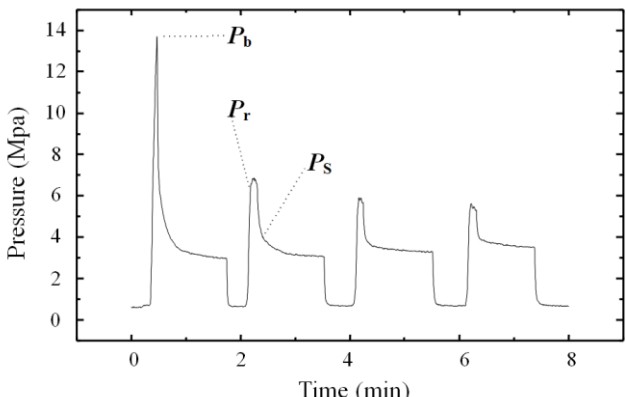

**Figure 2: Typical example of the pressure-time recording curve in hydraulic fracturing.**





According to the elastic theory and the PKN mechanical model, as shown in Figure 3, the fracturing pressure of the rock in the fracturing section is:

$$P_b = 3\sigma_h - \sigma_H + T \tag{5}$$


Among them, $\sigma_H$ and $\sigma_h$ are the maximum and minimum horizontal principal stresses, respectively, and T is the tensile strength of the rock. The fractures induced by hydraulic fracturing are vertical fractures and perpendicular to the direction of the minimum horizontal principal stress. Eq.(5) indicates that the fracturing pressure of rocks is independent of the size of the borehole and the elastic modulus of the rock, and is mainly determined by the tensile strength of the rock and the magnitude of the in-situ stress around the borehole.


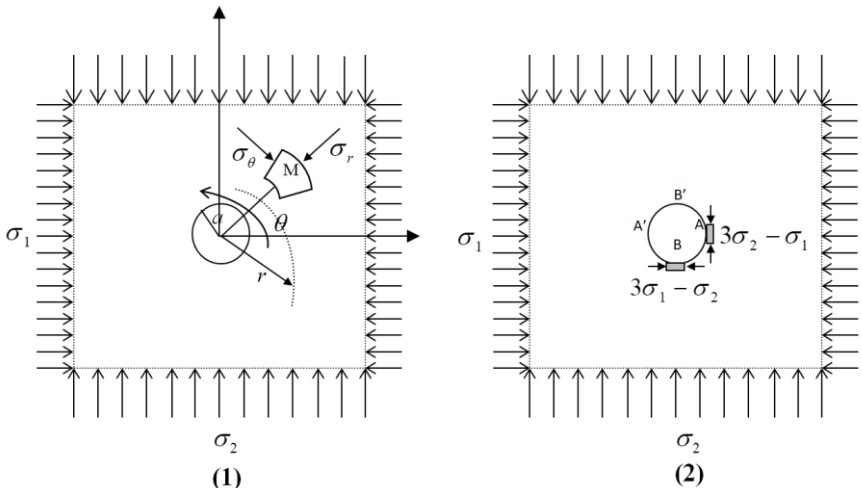

**Figure 3: Mechanical models of hydraulic fracturing measurement. (1) Stress acting on an infinitely large plate with a circular hole; (2) Stress concentration on the wall of a circular hole**

**2.3 Data processing methods**

From the pressure-time recording curve shown in Figures 2, the rock's fracture pressure $P_b$, instantaneous closure pressure $P_s$, and reopening pressure $P_r$ can be directly obtained. Based on these fundamental parameters, the maximum horizontal principal stress $\sigma_H$, minimum horizontal principal stress $\sigma_h$, and in-situ tensile strength of the rock T can be calculated.

(1) Fracture pressure $P_b$: The peak pressure of the first cycle in the hydraulic fracturing process is referred to as the fracture

pressure of the rock.

(2) Reopening pressure $P_r$: It refers to the pressure exerted when existing fractures reopen during subsequent pressure cycles. Typically, the point corresponding to a significant change in the slope of the pressure-time curve (Figure 2) is taken as the value of the reopening pressure of fractures.





(3) Instantaneous closure pressure $P_s$: The instantaneous closure pressure $P_s$ is equal to the minimum horizontal principal

stress $\sigma_h$. Therefore, the determination of the instantaneous closure pressure $P_s$ is crucial for hydraulic fracturing measurement, and the single-tangent method is commonly used to determine the value of $P_s$.

## 3 Research on key technologies of the small-diameter measurement system

The small-diameter hydraulic fracturing in-situ stress measurement system developed in this study is a dual-circuit hydraulic fracturing in-situ stress system. It utilizes a pair of expandable packers to isolate a section of the borehole at the selected

measurement depth. The system then applies pressure to this test section (commonly referred to as the fracturing segment) by pumping fluid, while simultaneously recording the pressure changes over time using a data acquisition system. The recorded data is then analyzed to obtain characteristic pressure parameters. Based on the stress calculation formulas described in Chapter 2, the values of the maximum and minimum horizontal principal stresses at the measurement point, as well as the hydraulic fracturing tensile strength and other rock mechanics parameters, can be determined.

As shown in Figure 4, the small-diameter hydraulic fracturing in-situ stress measurement system mainly consists of underground measurement components (series of small-diameter packers and dual-circuit connecting installation rods) and surface control components (hydraulic fluid control system, data acquisition and analysis system, and controllable flow high-pressure pump).

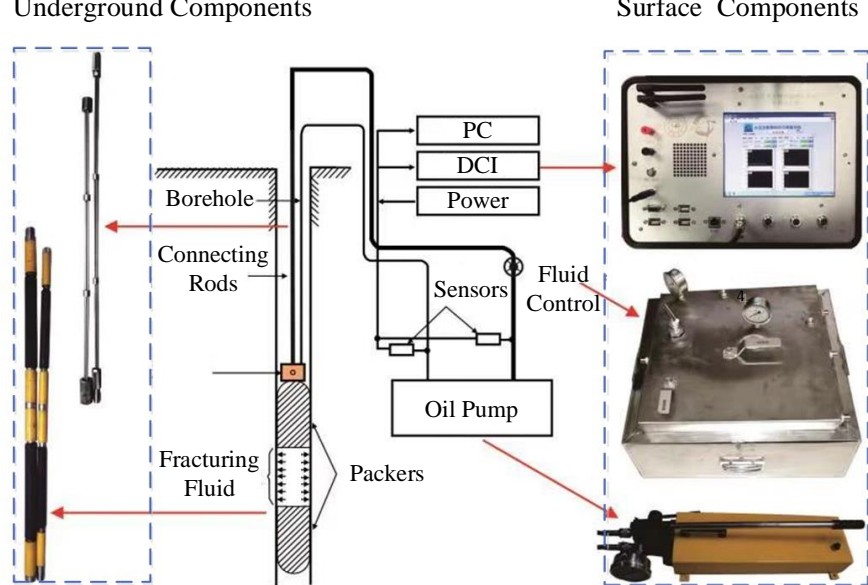

**Figure 4: Schematic diagram of the small-diameter hydraulic fracturing in-situ stress measurement system.**





## 3.1 Underground measurement components

The underground measurement components mainly consist of small-diameter packers and underground connecting installation rods. The function of the packers is to seal a section of intact rock wall in the borehole, forming a sealed space. The connecting installation rods are used in a multi-segment connection manner to connect the packers to the surface control

components at the borehole entrance.

### 3.1.1 Small-diameter packers

The packer is an essential tool used in oilfield casing or open-hole operations to isolate oil, gas, and water layers. It is a downhole tool used for layering in oil production engineering. Its main component is a rubber cylinder, which can be shortened in length and expanded in diameter through hydraulic or mechanical actions. This allows for the sealing of the

annular space, separating the upper and lower oil (gas, water) layers, thereby enabling layered testing, layered oil production, layered water injection, layered reformation, and water layer plugging in oil and water wells (Niu, 2013).

Traditional packers used in oil and gas fields are relatively large in size, with an outer diameter ranging from 90 to 360mm, and the sealing pressure can reach 30 to 40MPa. In this study, the packers are primarily designed for use in hydraulic fracturing testing, to seal a section of intact rock wall in the borehole, creating a sealed space and providing high-pressure

resistance and sealing functions for subsequent hydraulic fracturing operations. The system has developed three different diameter hydraulic fracturing packers: Φ20, Φ31, and Φ42mm. Their detailed technical parameters are shown in Table 2.

**Table2.** Table of parameters for hydraulic fracturing packers

| Small-diameter packers | Parameter name | Design target values/state |
|---|---|---|
| Outer diameter of Φ31mm | Outer diameter/maximum expansion diameter | 30±1mm/45mm |
| | Length | 1200mm |
| | Structure | Dual-circuit crossover design |
| | Maximum sealing pressure | 40MPa |
| Outer diameter ofΦ42mm | Outer diameter/maximum expansion diameter | 42±1mm/63mm |
| | Length | 1200mm |
| | Structure | Dual-circuit crossover design |
| | Maximum sealing pressure | 45MPa |
| Outer diameter of Φ20mm | Outer diameter/maximum expansion diameter | 20±0.5mm/30mm |
| | Length | 500mm |





| Structure | Single-circuit push-pull type |
|---|---|
| Maximum sealing pressure | 30MPa |

(1) Packers with diameters of Φ31mm and Φ42mm

The structural design of Φ31mm and Φ42mm small-diameter packers utilizes a dual-circuit crossover sealing-fracturing structure. This crossover packer consists of a dual-circuit installation rod connector, an upper packer, a connector with water jet holes, a lower packer, and a lower plug. The composition and structure of the upper and lower packers are identical. The schematic diagram of the internal structure is shown in Figure 5.

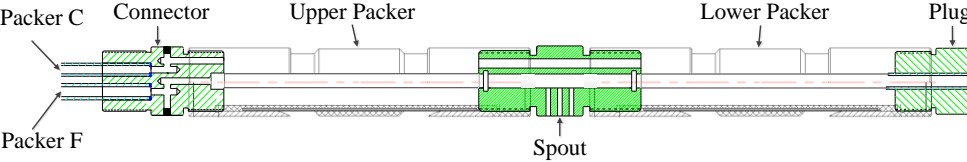

**Figure 5: Schematic diagram of the internal structure of the crossover packer.**

As shown in Figure 5, the two inlet holes of the dual-circuit installation rod connector supply water to both the packer and the fracturing segment. The water pressure fracturing test of the sealing segment is conducted through the water jet holes of the connector between the upper and lower packers. Simultaneously monitoring the pressure values of the fracturing segment and the packers helps obtain high-quality stress measurement data, enabling in-situ stress measurement in small-diameter

geological exploration boreholes ranging from Φ38mm to Φ50mm. The photo of the crossover packer is shown in Figure 6.

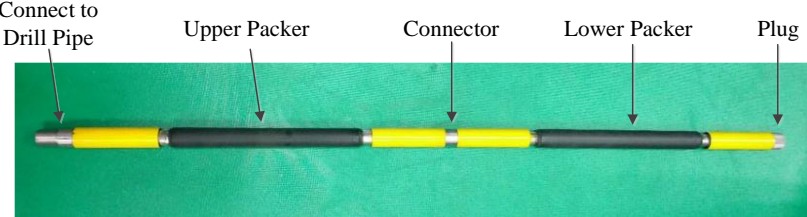

**Figure 6: Photo of the crossover packer.**

(2) Packer with a diameter of Φ20mm

The Φ20mm packer is designed with a single-circuit push-pull sealing-fracturing structure. This push-pull packer consists of

an upper connector, a sliding head, a limit connector, a rubber cylinder, and a lower connector. The schematic diagram of the overall internal structure is shown in Figure 7.




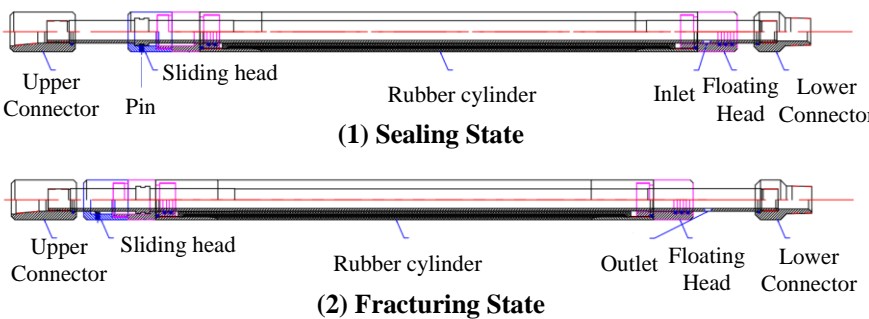

**Figure 7: Photo of the crossover packer.**

In Figure 7, (1) and (2) respectively represent the seat sealing state and fracturing state of the packer. After the packer is

placed in the target formation, it is pressurized through the injection port in the upper connector to the designed seat sealing pressure (4~5MPa). The liquid enters the rubber cylinder through the sliding head, upper connector, and flows through the upper cylinder, floating head, and lower connector to complete the expansion seat sealing of the upper and lower cylinders. After the seat sealing is completed, the pin installed in the sliding head is sheared off by pushing the installation drill rod, allowing the middle column of the packer to descend. The sliding head reaches the limit adapter, and at this point, the

fracturing fluid enters the test section through the injection port, central pipe, lower inlet hole, and injection hole until the fracturing rock wall is reached. After the completion of the formation test, the pump at the wellhead stops pressurizing, and the lifting column is raised to allow the upper injection fluid to return to its original position, releasing the pressure and unsealing the packer. After the cylinder completely retracts, the packer is lifted to the wellhead and secured with a pin at the sliding head, and then the testing continues in the next target formation.

**3.1.2 Dual-circuit connecting rod**

Due to the large diameter of conventional equipment such as drill rods and high-pressure hoses used in traditional hydraulic fracturing in-situ stress testing, the connecting installation rod of the small-diameter hydraulic fracturing measurement system adopts an integrated dual-circuit structure. It utilizes an internal end-face sealing form, with two high-pressure stainless steel pipes of the installation rod serving as separate fluid channels for the fracturing segment and the seat sealing

segment to provide high-pressure liquid. Two specifications (Φ30mm and 42mm) of small-diameter dual-circuit connecting installation rods are designed.

The small-diameter dual-circuit connecting installation rod consists of a dual-channel high-pressure circuit (314 stainless steel pipe), a male connector, two reinforcing ribs, a sleeve, and a female connector. The assembly diagram of the connecting installation rod is shown in Figures 8.





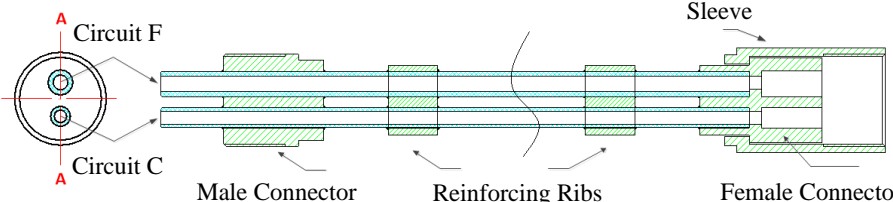


**Figure 8: Assembly diagram of the of the dual-circuit connecting rod.**

### 3.2 Surface Control Components

The surface control components consist of a high-pressure fluid control system and a controllable flow high-pressure oil pump. The high-pressure fluid control system is a key component for fluid extraction, transmission, and control during
hydraulic fracturing in-situ stress testing. The inlet of the high-pressure fluid control system is connected to the outlet of the high-pressure water pump, and the two outlet ports of the high-pressure fluid control system are respectively connected to the seat sealing pressure circuit and the fracturing pressure circuit of the connecting installation rod, achieving dual-circuit control of high-pressure fluid for hydraulic fracturing.

The high-pressure fluid control system includes a high-pressure oil pump, an oil tank, a pressurize ball valve, a relief ball
valve, a needle valve, a pressure gauge, a pressure sensor, a PC data acquisition and processing unit, and multiple three-way pipe fittings. It can sequentially perform high-pressure permeability self-checking and sensor calibration tests, seat sealing function test, high-pressure output function test of the fracturing segment, and pressure relief function test. This system realizes the integration and implementation of various high-pressure fluid control functions during the hydraulic fracturing in-situ stress measurement process.. Figure 9 shows the schematic diagram of the high-pressure fluid control system
connection.

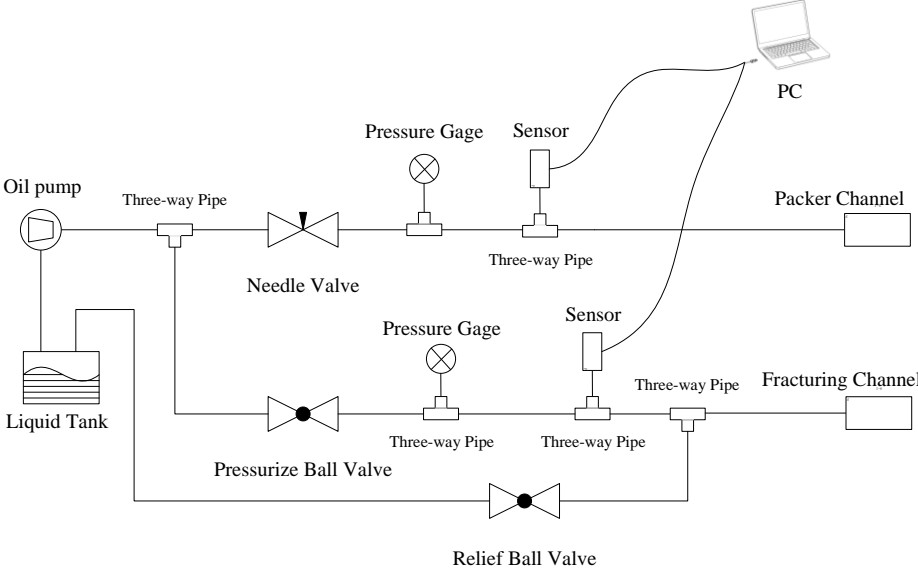





**Figure 9: Schematic diagram of connection of the high-pressure fluid control system.**

As shown in Figure 9, the high-pressure oil pump pumps the liquid from the oil tank to the three-way junctions. The three-way junctions are connected to the channels of the packer and the fracturing segment, respectively. In the packer channel, there are sequentially connected needle valves and 2 three-way junctions. The third channels of the two three-way junctions are connected to the pressure gauge and the pressure sensor, respectively. In the fracturing segment channel, there are sequentially connected pressurized ball valves and 3 three-way junctions. The third channels of the three-way junctions are connected to the pressure gauge, pressure sensor, and relief ball valve, respectively. The relief ball valve outputs to the oil tank via a pipeline. Both the sensors output to the PC data acquisition and processing unit.

## 4 The field measurement

### 4.1 Introduction to the measurement area

The Nalin River coal mine in Wushen Banner, Ordos, Inner Mongolia, belongs to the Nalin River mining area in the Dongsheng Coalfield. The stratigraphic division of this area falls within the Ordos subdivision of the North China Stratigraphic Region (Tian et al., 2011). The region, represented by Dongsheng —Wushen Banner, exhibits a relatively complete development of Mesozoic and Cenozoic strata, referred to as the "Wushen Banner subregion." The Nalin River exploration area is located in the southeastern part of the "Wushen Banner Subregion," as shown in Figure 10.

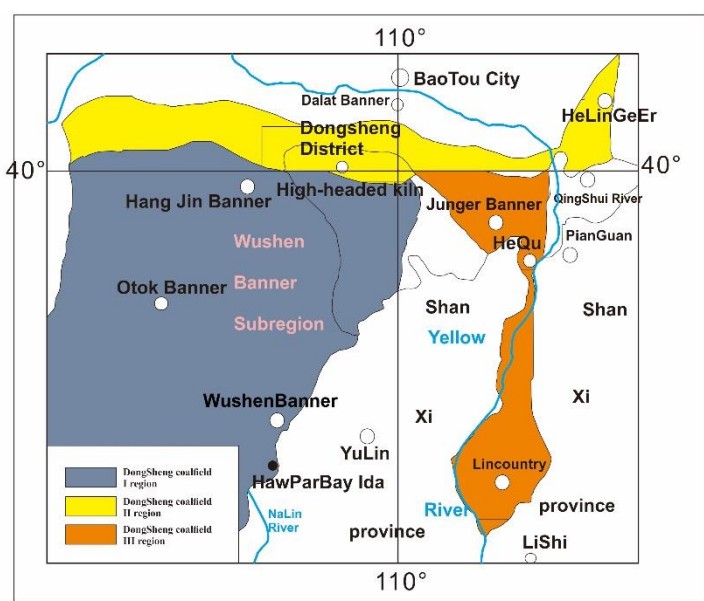

**Figure 10: Location Map of the exploration area.**

The Dongsheng Coalfield is located on the northeastern edge of the basin, while No.2 Well of the Nalin River mining area, is closer to the central zone of the basin. In the entire Ordos Basin, both in terms of basin genesis and coal accumulation



patterns, the Upper Triassic Yanchang Formation ($T_{3y}$) serves as the sedimentary basement for Jurassic coal-bearing basins and coal-bearing strata (Tian et al., 2011). Therefore, it is necessary to employ a small-diameter hydraulic fracturing in-situ stress measurement system for deep coal mining operations in order to conduct stress testing of the geological stress in coal seams. This will be beneficial in analyzing the characteristics and stability of the geological stress field in the coal mining

engineering area.

## 4.2 Measurement technology of small-diameter hydraulic fracturing

(1) Selection of the test section: Based on the depth position of the complete core recorded and verified, as well as the positions required by the engineering design, consideration should be given to placing the packer at a location with smooth borehole walls and consistent aperture.

(2) Inspection of the measurement system: Prior to the formal fracturing process, a leak test should be conducted on the drill rods and fracturing system used for testing, with the test pressure generally not lower than 15MPa. To ensure the reliability of test data, it is required that there be no leaks at any joints. Furthermore, the tested drill rods should be numbered to ensure accurate depth measurement.

(3) Installation of underground measurement components: Using dual-circuit connecting rods, a the crossover packer is

placed at the desired depth for measurement.

(4) Seat sealing: The crossover packer is pressurized by a water pump on the ground to expand and make close contact with the hole wall, forming a sealing space for the fracturing section.

(5) Hydraulic fracturing: Use a high-pressure pump to pressurize the test section through the connecting rods. During the pressurization process, the rocks in the test section will rupture at the position of the minimum tangential stress under

sufficient pressure, that is, in the direction perpendicular to the minimum horizontal principal stress. This pressure value will be recorded by the pressure sensor.

(6) Pump shut-off: After the rock fractures, the pump will be shut off, and the injection of pressure into the test section will be stopped. Under the action of rock stress and elasticity, cracks tend to close. The pressure recorded when the crack is in a critical closed state is the instantaneous closure pressure.

(7) Pressure release: When the pressure within the section stabilizes, the pressure within this section can be released, causing the opened cracks to close.

During the measurement process, each section usually undergoes 3-5 cycles in order to obtain reasonable stress parameters and accurately determine the fracture and extension status of the rock.

## 4.3 Analysis of measurement results

The stress values obtained from the three boreholes arranged in the engineering area are relatively consistent and increase with depth. The maximum and minimum horizontal principal stress values and vertical stress values were analyzed and





statistically analyzed, and the average values for each measurement interval were calculated. The statistical results are shown in Table 3.

**Table3.** Table of parameters for hydraulic fracturing packers

| Borehole identification | Depth range(m) | Maximum horizontal principal stress | | Minimum horizontal principal stress | | Vertical principal stress | |
|---|---|---|---|---|---|---|---|
| | | Range of stresses (MPa) | Mean value (MPa) | Range of stresses (MPa) | Mean value (MPa) | Range of stresses (MPa) | Mean value (MPa) |
| ZK1 | 533.3-547.4 | 13.20-22.99 | 19.58 | 6.94-11.79 | 10.10 | 13.87-14.22 | 14.04 |
| ZK2 | 533.1-548.3 | 16.16-21.54 | 18.72 | 8.37-11.06 | 9.60 | 13.87-14.25 | 14.06 |
| ZK3 | 534.1-548.1 | 11.54-21.14 | 18.03 | 6.05-11.05 | 9.21 | 13.91-14.24 | 14.06 |


Overall, within the testing depth range, the relative magnitudes of the three principal stresses can be described as follows: $S_H > S_V > S_h$. This indicates that in the shallow crust of the region, horizontal stresses play a dominant role, and the present-day stress conditions are relatively intense. Based on the general statistical analysis in Table 3, it can be determined that the range of in-situ stress values in the tested area should be as follows: the range of maximum horizontal principal stress values is 13.20 ~ 22.99 MPa, the range of minimum horizontal principal stress values is 6.05 ~ 11.79 MPa, and the range of vertical principal stress values is 13.87 ~ 14.25 MPa.

The relationship between the three principal stresses in this region is $S_H > S_V > S_h$, indicating a stress state favorable for slip on strike-slip faults. This result is consistent with the findings of Fan et al. (2003) using focal mechanism analysis, which suggest that seismic events in this region are primarily generated by slip on strike-slip faults. By conducting linear data fitting on the stress data, the distribution patterns of maximum, minimum, and vertical principal stresses with depth can be obtained (Figure 11), represented respectively as: $S_H = -0.28 + 0.035 \times Z$, $S_h = -0.29 + 0.020 \times Z$, $S_V = 0.023 \times Z$ (with correlation coefficients of 0.99, 0.98, and 1). It can be observed that both the maximum and minimum horizontal principal stresses increase with depth. The calculated stress gradients for the maximum and minimum stresses are 0.035 and 0.020, respectively, indicating that the stress state in this region is conducive to slip fault activity. The relatively small stress gradient for vertical stress is due to the presence of predominantly loose soil in the upper covering layer at the measurement point, reflecting an overall lower average density characteristic. Due to the excellent linear correlation of the data, these fitting formulas can be used to predict and estimate the stress state in other parts of the mining area.





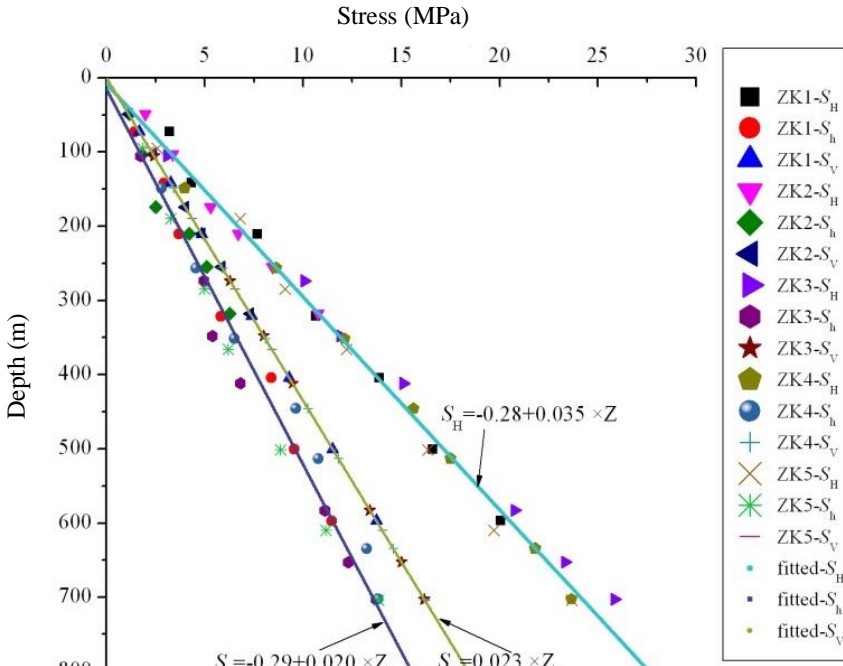

**Figure 11: Distribution characteristics of the maximum, minimum, and vertical principal stresses with depth.**

The data was plotted based on the depth distribution characteristics, resulting in a graph illustrating the lateral pressure coefficient, as shown in Figure 12. It can be observed that the lateral pressure coefficient exhibits a relatively scattered distribution, without displaying a common distribution pattern. Therefore, the arithmetic average of the lateral pressure coefficient was calculated for all the data.  The average ratio of the maximum horizontal principal stress to the vertical principal stress is 1.81, while the average ratio of the minimum horizontal principal stress to the vertical stress is 0.82.  The

average ratio of the maximum horizontal principal stress to the vertical principal stress at this measurement point is higher than the value of 1.5 obtained at the Wushenqi 3-1 coal mine stress measurement point.  This is because the measurement point includes more shallow data, which typically have higher measurement coefficient values. Overall, at a depth of 550m, the stress level at this measurement point is comparable to that of the Wushenqi 3-1 coal mine stress measurement point.





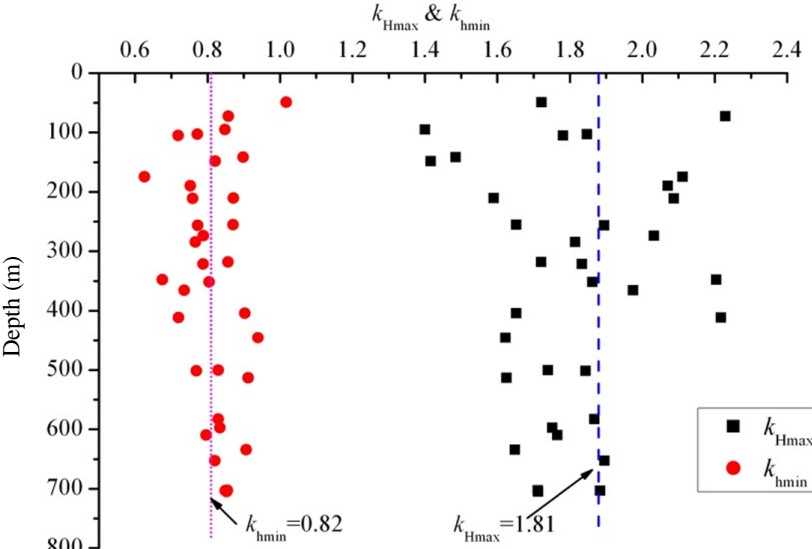

Figure 12: Distribution characteristics of the lateral stress coefficient with depth.

## 5 Conclusion

Observation and estimation of the stress state in the deep crust pose a significant challenge in field stress measurement. The hydraulic fracturing method is an important borehole-based technique for absolute stress measurement. The small-diameter hydraulic fracturing in-situ stress measurement system described in this study consists of underground measurement components (serial small-diameter packers and dual-circuit connection installation rods) and surface control components (hydraulic fluid control system, data acquisition system, and controllable flow high-pressure pump), enabling serial testing of small-sized boreholes for stress measurement.

The small-diameter hydraulic fracturing in-situ stress measurement system developed in this research possesses advantages such as simplicity in structure, portability, short test duration, high success rate, low requirements for rock integrity and pressurization equipment. It has been developed into a series of three types of small-diameter testing devices with diameters of Φ20, 31, and 42mm. This system fills the gap in the current domestic and international hydraulic fracturing in-situ stress measurement systems with diameters less than 45mm. Moreover, it has been innovatively applied and extended to the field of underground tunnel safety evaluation in coal mines and metal mining areas. It holds significant practical value and economic significance in advancing hydraulic fracturing in-situ stress measurement, promoting technological upgrades and innovations, and expanding the application scope of this technique.

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
