# Peer review of "Research and application of small-diameter hydraulic fracturing insitu stress measurement system"

_Geoscientific Instrumentation, Methods and Data Systems, 2023_

## Author Response (AR1)

**Reply to reviewer 1**

Dear reviewer 1#:

We are very grateful to this referee comments, and have carefully read and considered the referee's comments, and these comments are important for improving the quality of this manuscript. Based on these comments, we have made carefully modification and proofreading on the original manuscript, the revised parts have been marked in red in revised version, and the detail modifications are shown in next chapter.

Thank you very much for your suggestion and consideration, and we look forward to hearing from you.

Best regards,

Yimin Liu and Huan Chen.

Detailed revision:

(1) A more detailed discussion of the data obtained and how these results can be compared with other related research or applications in the field would be useful. It could be useful to add a more detailed discussion of its limitations and suggestions for future work.

Modification: Thanks a lot for the constructive suggestion. Based on measured data analysis, we have come to the following conclusion: the relationship between the three principal stresses in this region is $S_H > S_V > S_h$, indicating a stress state favorable for slip on strike-slip faults. Then we compared with other related research, such as the findings of Fan et al. (2003) using **focal mechanism analysis**, and Fan et al. also obtained corresponding conclusions that $S_H > S_V > S_h$, to support our conclusion and measurement accuracy.

According to the reviewer's comments, we have added the limitations of current research and suggestions for future work, mainly improve the theory of in-situ stress testing for small diameter hydraulic fracturing, especially the analysis of the effect of size effect on the fracture pressure value.

(2) The caption of Figure 9 is on the next page and Table 2 is split into two pages, it would be better to put them on the same page.

Modification: Thanks a lot for the kindness suggestion, we have put Figure 9 and Table 2on the same page.

(3) In general, it might be useful to check all captions, figures and equations (e.g. equation (5)) so that there is a good division with respect to the text.

Modification: According to the reviewer's comments, we have carefully checked captions and serial number of all figures, equations and tables.

(4) In addition, in some figures (e.g. Figures 1, 10 (legend), 11, 12) the captions

are not well visible.

Modification: Thanks a lot for this suggestion, we have redrew the Figures 1, 4, 5, 6, 10 (legend), 11 and 12, so that the legends and captions of these figuires are much more visible.

**Reply to reviewer 2**

Dear reviewer 2#:

On behalf of my co-authors, we thank you for giving us an opportunity to revise this paper, we appreciate editor and reviewers very much for their positive and constructive comments and suggestions on our manuscript. Based on these comments, we have made carefully modification and proofreading on the original manuscript. For the questions from reviewer 2#, I will explain in detail in the next chapter, and the detail modifications are also shown in red in revised version.

Thanks for your suggestions and comments. All your comments are very important, and they have important guiding significance for our future research work, and we look forward to hearing from you.

Best regards,

Yimin Liu and Huan Chen.

Detailed revision:

(1) The abstract section is relatively clear and concise, but I still suggest clarifying the differences of the new measurement system more clearly.

Modification: The Table1 shows measurement methods and technical parameters of the ready-made measurement devices, so there is a lack of hydraulic fracturing in-situ stress measurement system with a diameter smaller than 45mm. According to this comment, we have added the progressive and differences from the existent technology.

(2) Section2.3 show the data processing methods of hydraulic fracturing in-situ stress measurement, I think this is a principle based content that is not closely related to the characteristics of the article and can be deleted..

Modification: Thanks a lot for the kindness suggestion, we have deleted the Section2.3 that is not closely related to the main idea of this manuscript.

(3) The Figure 1 doesn't seem very clear, I suggest redrawing it. And the typeface in Figure 11, 12 seems too small, I suggest bolding and unifying. The same image and table should be displayed on the same page, suggest checking and correcting the entire text.

Modification: Thanks a lot for this suggestion. According to the reviewer's

comments, we have carefully checked captions and serial number of all figures, equations and tables. Due to our negligence, we have redrew the Figures 1, 4, 5, 6, 10 (legend), 11 and 12, so that the legends and captions of these figures are much more visible.

(5) The Section 6, I suggest further strengthening the induction and summary in Section conclusion.

Modification: According to this comment, we have reorganized and summarized, and added the research directions in subsequent research, mainly by applying the error compensation model of fluid mechanics influencing factors to the calculation and analysis of in-situ stress, improving the theory of in-situ stress measurement for inclined hole hydraulic fracturing, and especially analyzing the effect of size effect on the fracturing pressure value.